# Carbon Monoxide Oxidation over Gold Nanoparticles Deposited onto Alumina Film Grown on Mo(110) Substrate: An Effect of Charge Tunneling through the Oxide Film

**DOI:** 10.3390/ma14030485

**Published:** 2021-01-20

**Authors:** Tamerlan Magkoev

**Affiliations:** Laboratory of Surface Physics and Catalysis, Department of Condensed Matter Physics, North Ossetian State University, Vatutina 44-46, 362025 Vladikavkaz, Russia; t_magkoev@mail.ru; Tel.: +7-91-8822-4595

**Keywords:** surface reaction, oxide supported metal nanoparticles, carbon monoxide oxidation, electron tunneling, gold, alumina

## Abstract

Formation of gold nanosized particles supported by aluminum oxide film grown on Mo(110) substrate and oxidation of carbon monoxide molecules on their surface have been in-situ studied in ultra-high vacuum by means of Auger electron spectroscopy (AES), reflection-absorption infrared spectroscopy (RAIRS), low energy electron diffraction (LEED), atomic force microscopy (AFM), temperature-programmed desorption (TPD), and work function measurements. The main focus was to follow how the thickness of the alumina film influences the efficiency of CO oxidation in an attempt to find out evidence of the possible effect of electron tunneling between the metal substrate and the Au particle through the oxide interlayer. Providing the largest degree of surface identity of the studied metal/oxide system at different thicknesses of the alumina film (two, four, six, and eight monolayers), it was found that the CO oxidation efficiency, defined as CO_2_ to CO TPD peaks intensity ratio, exponentially decays with the oxide film thickness growth. Taking into account the known fact that the CO oxidation efficiency depends on the amount of excess charge acquired by Au particle, the latter suggests that electron tunneling adds efficiency to the oxidation process, although not significantly.

## 1. Introduction

Metal nanoparticles on oxides are used in a variety of applications, one of them being heterogeneous catalysis [1]. This motivates extensive studies of corresponding model metal/oxide supported catalysts to better understand the elementary steps involved in reactions on their surface [2]. One of the key points activating reactions on the surface of supported metal particles is the value of the charge that the particle acquires due to the charge transfer between the oxide support and the metal particle [3,4,5]. For instance, for Au/TiO_2_(110) Okazaki et al. [6] have shown via density functional theory (DFT) calculations that the electron transfer occurs from the six-fold Ti atom to the Au atom for the Ti-rich surface, while from the Au atom to the in-plane and inner oxygen atoms for the O-rich surface, which closely relates to the catalytic property of the Au/TiO_2_ system. A similar situation holds for Au supported on the other oxide—MgO, for which Sanchez et al. [7] have shown that there is partial electron transfer from the surface to the gold cluster, which plays an essential role in the activation of nanosize gold clusters as catalysts for the CO combustion reaction. Later on, Goodman and coworkers [8] also found magnesium oxide to be an effective electron donor to negatively charge supported Au particles via charge transfer from anionic vacancies (F-centers) and thus activating them as catalysts for CO oxidation. Catalytic activation of LiF supported Au particles via charge transfer from anionic vacancies has also been recently reported by Tvauri et al. [9]. Electron-rich Au nanoparticles are predicted to adsorb dioxygen more strongly and to activate the O-O bond via charge transfer from Au to form a superoxo-like species [10], as well as facilitating the activation of CO [11]. For metal particles deposited on nano-thick oxide layers grown on metallic substrates, there is an additional channel of charge transfer to/from the metal particles via electron tunneling through the oxide interlayer [12]. For example, deposition of Cs onto ultrathin alumina films supported on Mo(110) induces large (0.9–1.1 eV) positive shifts in the BE of O KVV, O 1s, Al 2p feature of alumina film [13]. Aluminum deposition on an alumina layer grown on NiAl(110) results in a shift of the oxide component in Al 2p, O 1s, and O 2p to higher BE by 0.47 ± 0.03 eV [14]. A similar shift to higher BE in the range of 0.5 eV was detected while depositing V onto the same oxide surface [15]. Considering that Cs, Al, and V have low work functions, a charge transfer is expected to take place from Mo to Cs [16], from NiAl to Al [14], and from NiAl to V [15]. These and similar studies mainly deal with the “static” state of supported particles, with no manifestation of how electron tunneling through oxide interlayer affects the chemical reaction on their surface. In relation to this, the aim of the present study was to find out the definite evidence of the effect of charge transfer to/from the metal particle via the electron tunneling through the supporting oxide film, grown on the conducting metal substrate, upon the possible chemical reaction over the supported particle. To minimize the effect caused by metal/oxide interaction, the noble metal (Au) and stable non-reducible oxide (Al_2_O_3_), as well as representative elementary reactants (CO + O_2_), are used.

## 2. Experimental

Investigations were carried out in a modified ultra-high vacuum (UHV) VG Escalab Mk II system (base pressure: 3 × 10^−10^ mbar) enabling Auger electron spectroscopy (AES) with the aid of a single-pass cylindrical mirror analyzer with a coaxial gun, low energy electron diffraction (LEED) using rear view four-grid optics, reflection-absorption infrared spectroscopy (RAIRS, Nicolet, Seattle, WA, USA) with grazing incidence and reflection infrared beams, temperature-programmed desorption (TPD) using quadrupole mass-spectrometer, work function (WF) measurements by Anderson’s method, and atomic force microscopy (AFM) with the aid of an Omicron Nanoprobe (Omicron GmbH, Hamburg, Germany) instrument. The p-polarized light used in RAIRS provides sensitivity to molecular vibrations normal to the surface plane. This is the case for diatomic polar molecules like CO, which adsorb in an upright or tilted geometry on most metallic and non-metallic substrates [17]. An aluminum oxide film of controlled thickness and structure was grown on Mo(110) substrate held at elevated temperature via a well-known procedure of reactive thermal evaporation of aluminum atoms from Knudsen cell in an oxygen ambient at a partial pressure of about 10^−7^ mbar [18]. The thickness of the grown alumina film was determined by the deposited amount of metallic aluminum by the well-known procedure described by Goodman and coworkers [18,19]. The Mo(110) support as a refractory metal was chosen as it enables high-temperature treatment of the films to achieve equilibration and because of the close lattice match with α Al_2_O_3_(1000) to get epitaxial films even at very low thickness. The film thickness was estimated within the error of about 15% by measuring the aluminum atom flux via quartz microbalance and the Mo MNV (188 eV) Auger intensity attenuation upon the oxide film growth, according to the procedure described by Goodman and coworkers [18,19]. The Al deposition flux was additionally controlled via the work function change of Mo(110) substrate upon aluminum submonolayer film growth [20]. The amount of deposited metallic aluminum was controlled by the deposition flux and time. The Au atoms were deposited on alumina films by thermal evaporation of bulk Au (purity: 99.9999%) via the Knudsen cell at substrates held at room temperature and then annealed to 500 K for 3 min to achieve equilibration. The Au coverage was determined by quartz microbalance and verified by substrate (O KVV, 502 eV) Auger signal attenuation. The surface concentration of Au atoms of 1.45 × 10^15^ cm^−2^ was considered to be equal to 1 MLE (monolayer equivalent) of Au. Commercially available carbon monoxide and oxygen of research-grade quality were controllably admitted into the vacuum chamber through UHV leak valves. It is considered that exposure of 1 L corresponds to 10^−6^ Torr × 1 s. The sample holder, on which Mo(110) crystal was mounted coaxially to quartz microbalance, enabled cooling the sample down to 90 K via attached liquid nitrogen reservoir and annealing to moderate temperatures (1000–1500 K) by direct current flow across the substrate, and to 2700 K by electron bombardment.

## 3. Results and Discussion

Taking into account that the electron tunneling probability through the oxide dielectric layer depends exponentially upon its thickness, the measurements for CO oxidation efficiency over Au/Al_2_O_3_/Mo(110) were done for different alumina interlayer thicknesses, namely two, four, six, and eight monolayers. As the surface reaction is very sensitive to the state of the surface, it was essential to ensure that the structural, electronic, and adsorption properties of alumina films of different thickness, as well as that of Au/Al_2_O_3_, are the same in order not to mask the possible tunneling effect on the surface reaction. It is well established by numerous studies that two monolayer thick alumina film exhibits properties that resemble those of bulk alumina [18,19,21,22,23]. Moreover, beginning from this thickness, the film acquires dielectric properties [24,25,26]. The LEED patterns of alumina films of two, four, six, and eight monolayer thickness are the same and demonstrate hexagonal symmetry corresponding to α-Al_2_O_3_(1000), which is the case for equilibrium alumina films on Mo(110) [18,19,27] (Figure 1a,b). Aluminum interatomic Auger peak position, as well as its relative intensity against oxygen O KVV Auger line, are essentially the same for different alumina films studied. This indicates that the stoichiometry of the films is similar. A more precise check of the surface state of alumina films is the adsorption behavior of test species, like CO [28]. One of the most sensitive tools to probe such behavior is RAIRS due to its high resolution and sensitivity to adsorbed molecules, of which the vibrational mode is extremely sensitive to details of the structural and electronic state of the adsorbent [17]. The IR spectra in the stretching region of CO adsorbed at saturation coverage (exposure: 200 L) on two, four, six, and eight monolayer thick alumina films cooled down to 90 K, are shown in Figure 2. As seen, the spectra are very similar, indicating that the state of alumina film surface at all thicknesses studied is essentially the same. Since it is known that CO does not adsorb on alumina regular sites at 90 K [29], the observed IR band is likely attributed to the CO bound to oxide defect sites [30]. The same IR band position, intensity, and halfwidth allow one to assume that the defect nature and their density are similar for all alumina films. The quite low IR intensity (signal to noise ratio) suggests that the density of defects is rather low. The most reasonable nature of these defects is anionic vacancies, which anyway exist to a certain extent in structurally ordered mostly stoichiometric alumina films grown by reactive deposition [31]. They enhance sticking of CO molecules to alumina surface by switching backdonation to CO 2π* antibonding orbital via the negative charge of F-center, which otherwise is negligible at regular α-Al_2_O_3_(1000) sites [32].

Morphology of Au overlayers, deposited in an equivalent amount of 0.7 MLE onto alumina films of different thicknesses held at 500 K to achieve equilibration, demonstrates similarity, as seen from representative AFM images (Figure 1c,d). This is confirmed by the similarity of the Auger Au NVV to Al LVV intensity ratio (Figure 1e,f)—the value characterizing the relative area of the substrate covered by the adsorbate [19,33]. According to the DFT calculations of Jennison and coworkers [34,35], the Au bounds to α-Al_2_O_3_(1000) by ionic bond at submonolayer coverage, whereas at high coverage, when 3D islands form, the metal to oxide bonding is dominated by polarization effect. The feature of Au deposits for oxide supports is that they are basically neutral on the most stable adsorption sites [12,36,37]. To more precisely check whether the state of Au particles depends on alumina film thickness, the adsorption of CO molecules as a test specie was probed by RAIRS (CO exposure: 100 L). All spectra consisting of a CO stretching vibrational band at 2098 cm^−1^ are almost identical to each other (Figure 3). Taking into account an extreme sensitivity of molecular vibrational modes to the details of the atomic and electronic structure of the substrate [17], the observed similarity of IR spectra points to the identical morphological and structural character of Au overlayer regardless of the underlying alumina film thickness. Since CO hardly adsorbs on the regular Au surface for the conditions used [38], the observed IR band can be attributed to molecules bound to undercoordinated sites available at Au clusters, as well as to the metal/oxide perimeter interface, as follows from numerous interpretations made for this and other types of oxide supported Au particles [12,29,38,39,40]. The lower CO stretching wavenumber on Au particle (2098 cm^−1^) compared to that on alumina support (2115 cm^−1^) can be rationalized in terms of larger charge backdonation from metal particle to CO 2π* antibonding orbital than from the anionic vacancy site of the oxide.

Postadsorption of oxygen at an exposure of 100 L on CO/Au/Al_2_O_3_ held at 90 K leads to a blue shift of CO IR band by 5–6 cm^−1^ with no noticeable effect on its intensity. This rather small wavenumber shift is hardly due to displacement of CO to another adsorption site upon oxygen postadsorption. The latter would have been characterized by a notably higher molecular vibrational shift [41]. As it is known that molecular oxygen dissociates on Au ultrasmall particles [3], the observed IR blue shift can be accounted by CO 2π* antibonding orbital depopulation via partial charge transfer to higher electron-affinity atomic O 2p level [42]. The observed quite low sensitivity of CO IR spectra to O_2_ postadsorption suggests that carbon monoxide and oxygen do not compete for the adsorption site on the supported gold particle at 90 K. Instead, they likely reside at the adjacent sites, slightly affecting each other as manifested by the indicated CO IR band shift. According to DFT calculations [43], a hollow site consisting of four Au atoms in a square geometry appears well suited for atomic oxygen, whereas CO primarily bounds to atop Au cluster undercoordinated sites. A more pronounced effect ultimately resulting in molecular conversion (CO + O→CO_2_) appears upon heating to facilitate the migration of the adsorbed species across the surface and their activation. The TPD spectra of CO and CO_2_ species, acquired during continuous exposure of the CO/Au/Al_2_O_3_ to oxygen by backfilling the UHV chamber to a partial pressure of 10^−6^ mbar are shown in Figure 4a,b. The temperature sweep rate was chosen as low as 1 K/s to ensure equilibration of the process, and the TPD mass-spectrometer was adjusted for simultaneous registration of CO (m/z = 28) and CO_2_ (m/z = 44) signals within one span. As seen, along with the desorption of CO (Figure 4a), the CO_2_ is also formed, although to a lower extent (Figure 4b). The mechanism of CO oxidation over oxide supported Au particles is under extensive study for the last several decades. It is considered to be a combination of a variety of parameters, such as particle size, morphology and structure, nature of the oxide support and its stoichiometry, state of the metal/oxide perimeter interface, reaction condition, etc. [12,29,38,39,40,44]. The common view in this regard is that the sign and magnitude of the charge of the Au particle, controlling activation of CO and O_2_ species, is a crucial point that determines the reaction pathway and rate [44].

Closer inspection of the TPD peak maxima (Figure 4, inlays) reveals that there is a definite feature that the CO signal grows and the CO_2_ signal concomitantly drops as the oxide film thickness increases. It means that the efficiency of CO oxidation is higher for thinner alumina interlayer film. This efficiency can be qualitatively defined as the ratio of CO_2_ to CO TPD peak intensity. The corresponding plot versus the alumina film thickness is presented in Figure 5. The plot quite well fits the exponential decay curve. The corresponding curve is plotted only to guide the eye as the precise thickness of the alumina film in the range from 0 to 8 mL is not unambiguously known. Taking into account the above-mentioned fact that the CO oxidation rate over supported Au particles depends on the value of their charge, the observed plot can be viewed as evidence that the tunneling of electrons of Mo(110) metal support through the alumina film to/from the Au/Al_2_O_3_ reaction interface adds to the efficiency of CO oxidation. The reaction requires an excess charge, which is withdrawn/donated from/to the underlying conducting support via switching of tunneling effect through the insulating oxide interlayer. Charging of Au particles needed to activate the carbon monoxide oxidation steps “pumps” electrons from/to the metal support via the tunneling. This is in line with DFT studies indicating that the charging/discharging of the cluster during the catalytic cycle of CO oxidation strongly influences the energetics of all redox steps in catalytic conversions [45]. Also, according to previously reported results [46], the O_2_ upon adsorption gains approximately one electron, O–O bond length elongates to 1.39–1.47 Å, depending on if the cluster size and the O–O bond length correlates with the charge state of the molecule. Another point is that the charge rearrangement induced by the adsorbates occurs at the metal/oxide interface, as predicted via calculations [45], requires an electron reservoir, which in the present case is the underlying metal support, supplying charge via tunneling through the oxide film. The latter effect, however, is not dominant. As seen in Figure 5, the CO oxidation efficiency gain is not significant and is within c.a. 20%.

Taking into account the possibility of electron tunneling to/from the Au particles through the oxide layer, one can expect that the charging of Au particles would be manifested by CO IR frequency shift with the oxide film thickness. The latter, however, is not the case. As seen in Figure 3, the corresponding CO stretching frequency is essentially the same regardless of the alumina film thickness. It means that in “static” mode the tunneling may be negligible. The effect mounts under reaction conditions when molecular oxidation requires extensive dynamical charging/recharging of the Au particle, the metal/oxide interface, and the reactants.

## 4. Conclusions

The structural, morphological, and adsorption properties of the Au overlayer on the ordered alumina film grown on Mo(110) substrate are similar regardless of the oxide film thickness, at least, in the range of two to eight monolayers. Unlike the regular surface of bulk Au, the CO readily adsorbs on the alumina supported gold clusters held at a temperature of 90 K, and presumably at the metal/oxide perimeter interface. This CO layer is quite slightly affected by the oxygen, dosed by backfilling the UHV chamber by O_2_ gas, suggesting that CO and oxygen occupy different adsorption sites. Heating of the formed (CO + O) adlayer results in desorption to the gas phase of both CO and CO_2_, the latter to a notably lower extent. This effect has a definite feature that the relative amount of desorbing CO_2_, compared to that of CO, exponentially drops as the alumina film thickness increases. The latter, on account of the known fact that the CO oxidation efficiency depends on the amount of excess charge acquired by the Au particle, metal/oxide interface, and the reactants, is evidence that electron tunneling between the reaction area and the metal support through the oxide interlayer stimulates the oxidation process.

## Figures and Tables

**Figure 1 materials-14-00485-f001:**
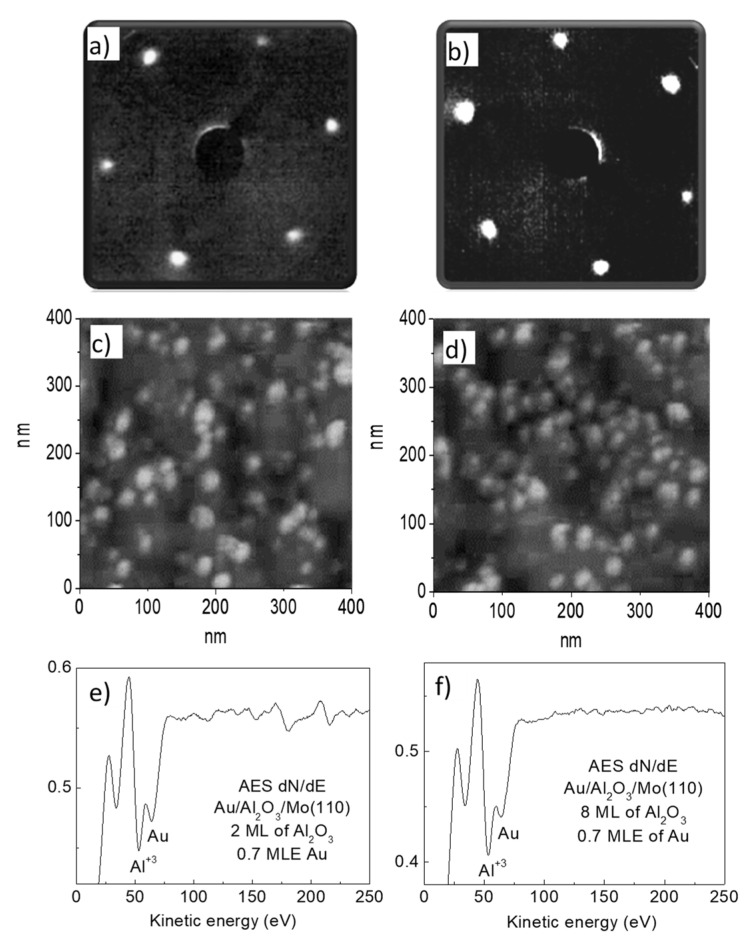
LEED patterns of alumina 2 mL (**a**) and 8 M (**b**) thick films on Mo(110) substrate. AFM images of Au overlayers at a coverage 0.7 MLE on alumina 2 mL (**c**) and 8 M (**d**) thick films. Gold was deposited onto alumina held at room temperature and then annealed to 500 K for 3 min. Auger spectra corresponding to AFM images: (**e**)→(**c**), (**f**)→(**d**).

**Figure 2 materials-14-00485-f002:**
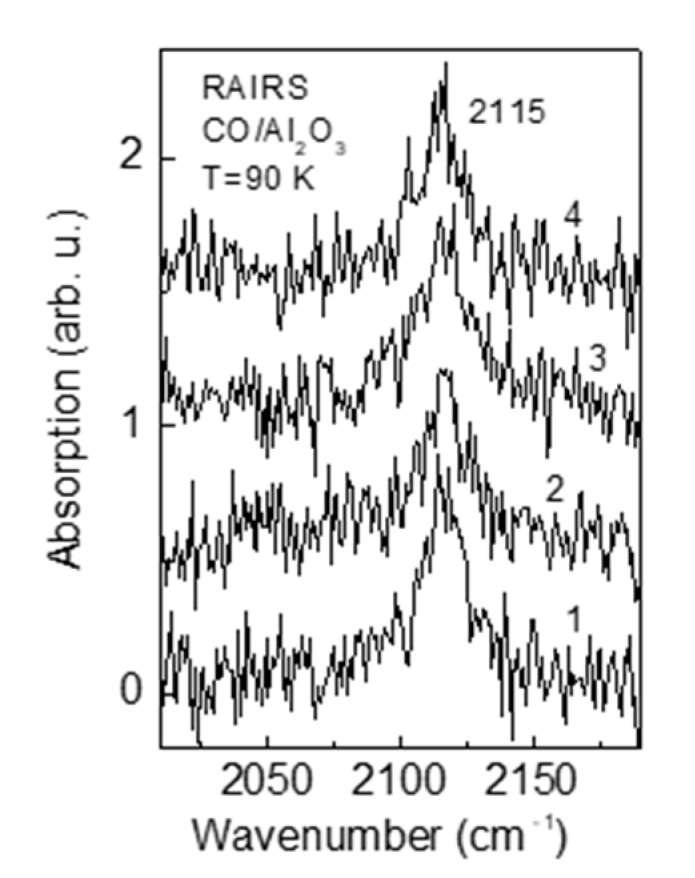
RAIR spectra in CO stretch region, obtained after 200 L-exposure of CO of alumina films 2 mL (1), 4 mL (2), 6 mL (3), and 8 mL (4) thick, held at a temperature of 90 K.

**Figure 3 materials-14-00485-f003:**
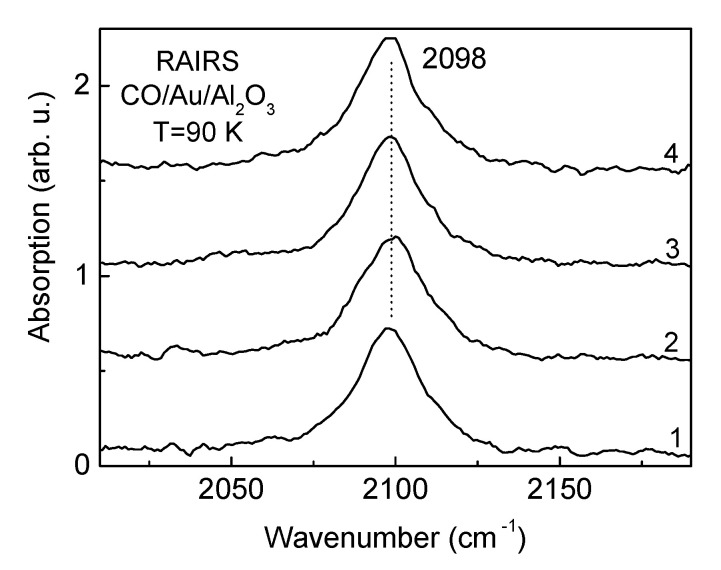
RAIR spectra in CO stretch region, obtained after 100 L-exposure of CO of alumina supported Au overlayers at a coverage of 0.7 MLE, cooled down to 90 K. Alumina film thickness: (1)—2 mL; (2)—4 mL; (3)—6 mL; (4)—8 mL.

**Figure 4 materials-14-00485-f004:**
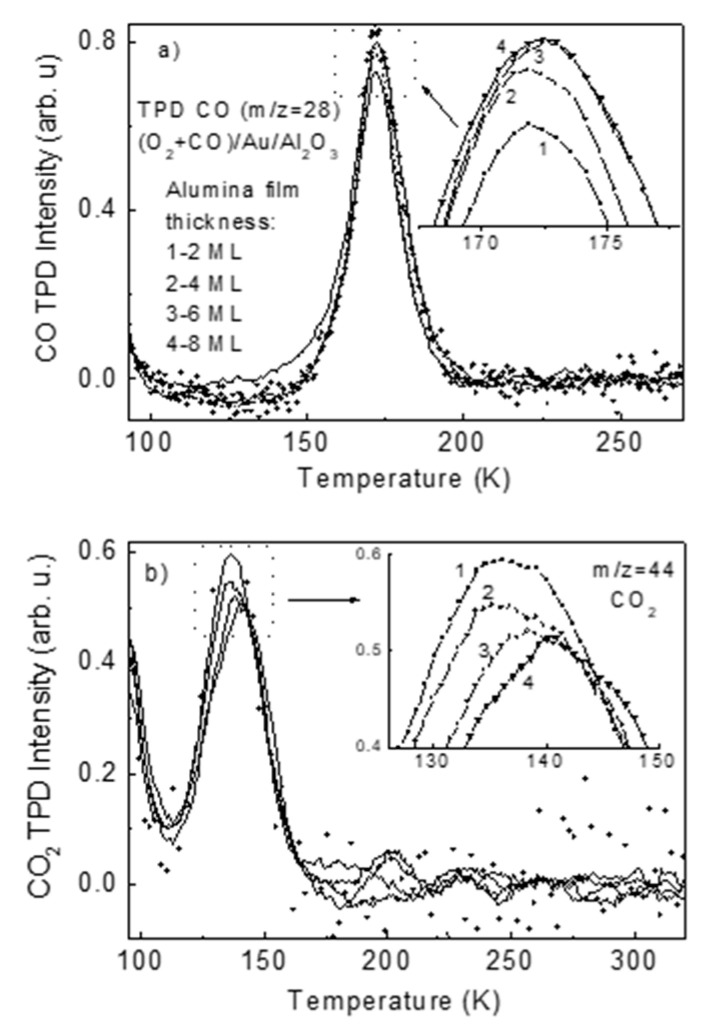
TPD spectra of CO (**a**) and CO_2_ (**b**), acquired during continuous exposure of CO/Au/Al_2_O_3_ to oxygen via backfilling the UHV chamber by O_2_ to a partial pressure of 10^−6^ mbar. Inlays depict the TPD peak maxima area. Alumina film thickness: (1)—2 mL; (2)—4 mL; (3)—6 mL; (4)—8 mL.

**Figure 5 materials-14-00485-f005:**
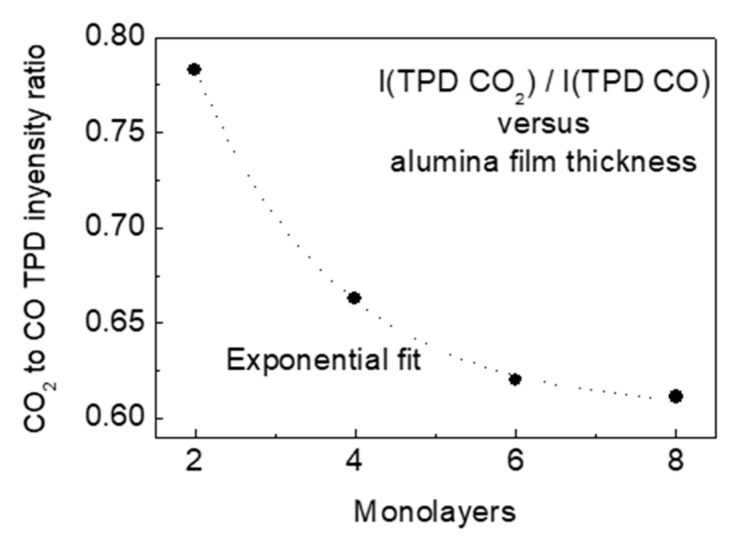
Ratio of TPD peak intensity of CO_2_ to CO versus the alumina film thickness (points) and the exponential fit (dotted line).

## Data Availability

Data available on request due to privacy restrictions. The data presented in this study are available on request from the corresponding author. The data are not publicly available due to institutional internal regulations.

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
