# Peer review of "Carbon Monoxide Oxidation over Gold Nanoparticles Deposited onto Alumina Film Grown on Mo(110) Substrate: An Effect of Charge Tunneling through the Oxide Film"

_materials, 2021, doi:10.3390/ma14030485_

Round 1

Reviewer 1 Report

The paper deals with the formation of gold nanosized particles supported by aluminum oxide film grown on Mo(110) substrate and oxidation of carbon monoxide molecules on their surface. The authors studied in-situ in ultra-high vacuum by means of Auger electron spectroscopy (AES), reflection-absorption infrared spectroscopy (RAIRS), low energy electron diffraction (LEED), atomic force microscopy (AFM), temperature programmed desorption (TPD) and work function measurements. They investigated how the thickness of the alumina film influences the efficiency of CO oxidation in an attempt to find out an evidence of possible effect of electron tunneling between the metal substrate and the Au particle through the oxide interlayer.

The paper is well organized, the authors used special investigation techniques, the paper is useful for the readers working in this area. 

Author Response

Dear Reviewer,

Thank you very much for reviewing the manuscript, the kind report and the recommendation.

Sincerely yours,

T. Magkoev

Reviewer 2 Report

Your manuscript entitled "Carbon monoxide oxidation over gold nanoparticles deposited 2 onto alumina film grown on Mo(110) substrate: An effect of 3 charge tunneling through the oxide film" is interesting to be published in Materials. However, prior the publication, I recommend minor revisions as following:

  1. On page 2. line 76, you mentioned the method to grow Alumina film. However, you did not explain how you can control the number of monolayers in details (time of deposition etc.). 
  2. On page 3, line 103, you mentioned the samples have 2, 4, 6, and 8 monolayers. However, the characterizations are from LEED patterns and quartz microbalance. Is it possible to have other measurements, e.g. AFM of the scratch region to measure the thickness?
  3. Several abbreviations need to be explained when they are first appeared; DFT (page 1, line 33), MII (page  2, line 67). 
  4. In Figure 5, you did the exponential fit to the TPD intensity ratio against monolayers. However, since you did not give the actual thickness values, it is not easy to derive the physics from this analysis. If it has no meaning, it is better to specify the fit as only to guide the eye.

Author Response

Dear Reviewer,

Thank you very much for reviewing the manuscript and the useful comments.

Below please find point-by-point description of changes made to the manuscript according to the comments:

Comment: On page 2. line 76, you mentioned the method to grow Alumina film. However, you did not explain how you can control the number of monolayers in details (time of deposition etc.).

Reply: The thickness of the grown alumina film was determined by the deposited amount of metallic aluminum by the well-known procedure, described in references [18,19] and was controlled by the metal deposition flux and time. Corresponding additions are made to manuscript (lines 79-81, 89-90).

Comment: On page 3, line 103, you mentioned the samples have 2, 4, 6, and 8 monolayers. However, the characterizations are from LEED patterns and quartz microbalance. Is it possible to have other measurements, e.g. AFM of the scratch region to measure the thickness?

Reply: The thickness of the films was determined via the absolute amount of deposited material, its growth mode and morphology. However, there are other ways to probe the film thickness, as suggested, by AFM or nanoindentation, or some other local tool. In these cases one can encounter a number of problems relating to tip and scratch proper alignment, low difference of contrast between the scratch side and the undelaying support and some others. These require a special care to take, which was beyond the main focus of the present work. Needless to say, this would add useful information regarding the precise value of the film thickness and its lateral variation at nanoscale for possible future studies.

Comment: Several abbreviations need to be explained when they are first appeared; DFT (page 1, line 33), MII (page  2, line 67).

Reply: Abbreviation is explained in the text (line 33). In line 67 the term “MII” was, unfortunately, misprint, and is replaced by “MkII” that denotes the model of   Escalab apparatus used.

Comment: In Figure 5, you did the exponential fit to the TPD intensity ratio against monolayers. However, since you did not give the actual thickness values, it is not easy to derive the physics from this analysis. If it has no meaning, it is better to specify the fit as only to guide the eye.

Reply: Actually, we meant the same, as the exact absolute thickness of the films is not unambiguously known. So, this accent is strengthened in the text (lines 209-211).

Sincerely yours,

T. Magkoev

Reviewer 3 Report

This is an interesting study and the manuscript is well written. I recommend publication of this manuscript after minor revisions. Followings my concerns:

1) Since the particle size of Au may affect the catalytic performance. It is possible to use other experiment or tool to further confirm that the Au deposited on alumina films of different thickness have very similar dispersion?

2) Is there any way to further verify there is electrons tunneling of Mo(110) metal support through the alumina film to/from the Au/Al2O3 reaction interface?

Author Response

Dear Reviewer,

Thank you very much for reviewing the manuscript and the useful comments.

Below please find point-by-point response to the comments:

Comment: 1) Since the particle size of Au may affect the catalytic performance. It is possible to use other experiment or tool to further confirm that the Au deposited on alumina films of different thickness have very similar dispersion?

Reply: One possible tool to directly probe the Au particle size and dispersion would be high-resolution scanning electron microscopy. In this case it should be in-situ tool to adequately compare SEM and AFM results. This would certainly add unambiguity to our results, however, we had no this tool at our disposal.

Comment: 2) Is there any way to further verify there is electrons tunneling of Mo(110) metal support through the alumina film to/from the Au/Al2O3 reaction interface?

Reply: One relevant way would be comparison of results obtained in this study with those obtained for the above Au/Al2O3 system, but with some highly insulation support (for instance, SiO2, BN) instead of metallic Mo(110), when no charge transfer from this insulating support is expected. However, this is a separate quite comprehensive study, which was beyond the scope of the present work. Nevertheless, this is certainly an attractive idea for possible future comparative study of conducting and non-conducting support in terms of further elucidation of charge tunneling effect upon the catalytic efficiency.

Sincerely yours,

T. Magkoev